# miR-29a-3p Regulates Autophagy by Targeting Akt3-Mediated mTOR in SiO_2_-Induced Lung Fibrosis

**DOI:** 10.3390/ijms241411440

**Published:** 2023-07-14

**Authors:** Peiyuan Li, Xiaohui Hao, Jiaxin Liu, Qinxin Zhang, Zixuan Liang, Xinran Li, Heliang Liu

**Affiliations:** 1School of Public Health, North China University of Science and Technology, Tangshan 063210, China; pyli2018@163.com (P.L.); liujiaxin19981021@163.com (J.L.); a15033963732@163.com (Q.Z.); lzx200107141108@163.com (Z.L.); lxr13363219912@163.com (X.L.); 2Hebei Key Laboratory of Organ Fibrosis, North China University of Science and Technology, Tangshan 063210, China

**Keywords:** silicosis, lung epithelial cell, autophagy, miRNA microarray analysis

## Abstract

Silicosis is a refractory pneumoconiosis of unknown etiology that is characterized by diffuse lung fibrosis, and microRNA (miRNA) dysregulation is connected to silicosis. Emerging evidence suggests that miRNAs modulate pulmonary fibrosis through autophagy; however, its underlying molecular mechanism remains unclear. In agreement with miRNA microarray analysis, the qRT-PCR results showed that miR-29a-3p was significantly decreased in the pulmonary fibrosis model both in vitro and in vivo. Increased autophagosome was observed via transmission electron microscopy in lung epithelial cell models and lung tissue of silicosis mice. The expression of autophagy-related proteins LC3α/β and Beclin1 were upregulated. The results from using 3-methyladenine, an autophagy inhibitor, or rapamycin, an autophagy inducer, together with TGF-β1, indicated that autophagy attenuates fibrosis by protecting lung epithelial cells. In TGF-β1-treated TC-1 cells, transfection with miR-29a-3p mimics activated protective autophagy and reduced alpha-smooth muscle actin and collagen I expression. miRNA TargetScan predicted, and dual-luciferase reporter experiments identified Akt3 as a direct target of miR-29a-3p. Furthermore, Akt3 expression was significantly elevated in the silicosis mouse model and TGF-β1-treated TC-1 cells. The mammalian target of rapamycin (mTOR) is a central regulator of the autophagy process. Silencing Akt3 inhibited the transduction of the mTOR signaling pathway and activated autophagy in TGF-β1-treated TC-1 cells. These results show that miR-29a-3p overexpression can partially reverse the fibrotic effects by activating autophagy of the pulmonary epithelial cells regulated by the Akt3/mTOR pathway. Therefore, targeting miR-29a-3p may provide a new therapeutic strategy for silica-induced pulmonary fibrosis.

## 1. Introduction

Pulmonary fibrosis (PF) is a lung disease characterized by the progressive and irreversible destruction of normal lung architecture. It has numerous causative factors, including respiratory infections and environmental exposure [1,2]. Silicosis is one of the most severe lung fibrotic diseases caused by the inhalation of crystalline silica dust in the environment for an extended period and is usually associated with occupations such as construction and mining [3,4]. It is one of the leading occupational diseases in many developing countries, including China [5], and poses a serious threat to public health as it remains incurable [6]. Therefore, new effective treatments and medications are urgently required to alleviate the progression of silicosis and reduce mortality in its late stages [7].

Several studies have found that autophagy is essential in the pathology of PF, including silica-induced PF [8,9]. In addition, epithelial cell dysfunction, including impaired autophagy, is a central component of the pathology of some lung diseases [10]. However, the molecular mechanism of how epithelial autophagy affects lung fibrosis remains to be elucidated. The pro-fibrotic cytokine transforming growth factor beta 1 (TGF-β1) induces epithelial-mesenchymal transition (EMT), promoting fibrosis [11,12]. In this study, fibrotic cell models were constructed with TGF-β1, and we focused on autophagy changes in epithelial cells and elucidated the underlying mechanism of how autophagy dysfunction affects silicosis progress in lung epithelial cells.

It has been proven that silica regulates autophagy activity via the Akt/mTOR signal pathway [13]. Akt is a central node of many signaling pathways, and its aberrant expression is a pathophysiological property of various diseases [14,15]. The Akt family, including the isoforms Akt1, Akt2, and Akt3 [16], are encoded by three independent genes that are not functionally complementary [17,18]. Akt1 and Akt2 are involved in multiple physiological processes, which include cell survival, proliferation, and differentiation [19,20]; however, the specific role of Akt3 remains largely unexplored. Akt3 plays an important role in some diseases. Studies have shown that silencing Akt3 can partially reverse LPS-induced acute lung injury [21]; however, few roles of Akt3 have been reported in silicosis. In this research, we investigated the mechanism of Akt3-mediated signaling pathways in silicosis.

MicroRNAs (miRNAs) are small non-coding RNAs that affect gene expression at the post-transcriptional level [22] and have been revealed as potential disease modifiers in various environmental respiratory diseases [23,24]. Several studies have shown that miRNAs are crucial for the emergence of many fibrotic diseases, including silicosis [25]. miR-29 has been reported to be significantly reduced in some fibrotic diseases associated with the lung, heart, and liver [26,27]. Nevertheless, the therapeutic implications of miR-29 in silicosis remain unclear.

In the current study, we investigated the function and potential mechanisms of miR-29a-3p in PF caused by silica. We found that miR-29a-3p was downregulated in silicosis models both in vivo and in vitro. Notably, miR-29a-3p overexpression not only increased autophagy levels but also partially prevented fibrosis in TGF-β1-treated lung epithelial cells. Additionally, we identified Akt3 as a target of miR-29a-3p. Our findings showed that miR-29a-3p promotes autophagy via the Akt3-mediated mTOR signaling pathway to suppress PF, revealing a possible therapeutic target for the treatment of silicosis.

## 2. Results

### 2.1. miR-29a Expression Is Significantly Downregulated in Fibrotic Lung Tissue Based on Bioinformatics Analysis

miRNA expression is disrupted in PF. To determine whether alterations in miRNA expression contribute to the phenotypes, we examined the differentially expressed miRNAs in the lungs of patients with PF and considered their intersection. According to GEO2R online analysis, 39 miRNAs with significantly differing expression levels were identified (*p* < 0.05) in GSE45789, of which 11 and 28 were highly and weakly expressed, respectively. In GSE32538, the expressions of 204 miRNAs were significantly altered (*p* < 0.05), of which 50 and 154 were highly and weakly expressed, respectively. Based on Refseq and Ensembl miRNA annotations, we identified 11 overlapping miRNAs, namely two and nine highly and weakly expressed miRNAs, respectively (Figure 1A). Among the nine weakly expressed miRNAs, miR-29a-3p attracted our attention. The miRDB database (http://mirdb.org, accessed on 10 March 2023) was used to identify the predicted targets of miR-29a-3p. The predicted target genes of miR-29a-3p were primarily associated with cancer-related pathways and the PI3K-Akt signaling pathway according to KEGG pathway analysis (Figure 1D). The results indicate that miR-29a-3p is closely associated with Akt-related signaling pathways. Akt3, as a predicted target gene of miR-29a-3p, is an isoform of the Akt family. Akt was found to promote fine particle-induced lung fibrosis through the regulation of autophagy [28]. Based on the above data, we hypothesized that miR-29a-3p may attenuate silica-induced pulmonary fibrosis through Akt3-mediated autophagy.

### 2.2. miR-29a-3p Expression Was Downregulated In Silica-Induced PF Model

In the miR-29 family, miR-29a-3p showed the most significant changes and was, therefore, the focus of the in vitro and in vivo studies to explore therapeutic approaches for the treatment of PF. The histopathological examination revealed significant parenchymal alterations in lung tissue after 7 and 14 days of silica exposure, including interstitial thickening, partial destruction of alveolar walls leading to alveolar fusion, and increased inflammatory cells observed around small airways, and after exposure for 28 days, diffuse infiltration of inflammatory cells and massive deposition of collagen fibers, with typical diffuse pulmonary fibrosis and silicotic nodules (Figure 2A). Western blot analysis showed that increasing silica exposure time markedly raised the fibrosis markers collagen I and α-SMA (Figure 2B) in agreement with the histopathological alterations. Subsequently, results from qRT-PCR revealed that lung tissues of mice with silicosis had significantly lower levels of miR-29a-3p than those in the control group mice (Figure 2C). TGF-β1 is a known fibrosis-promoting factor, which is also considered to be associated with silicosis [29]. Mouse lung epithelial cells (TC-1) were exposed to various concentrations of TGF-β1. According to the Western blot data, collagen I and α-SMA expression gradually increased as TGF-β1 concentration rose (Figure 2D), indicating that TC-1 cells acquire a mesenchymal phenotype with increased ECM. Based on the above results, 10 ng/mL of TGF-β1 was used as a positive control for the TC-1 cells model. Hence, all follow-up experiments were carried out at pH 7.4. Furthermore, the qRT-PCR results demonstrated that the level of miR-29a-3p decreased in a dose-dependent manner with increasing of TGF-β1 concentration (Figure 2E). In summary, our studies suggest that the miR-29a-3p was downregulated both in vitro and in vivo in the silica-induced PF model.

### 2.3. Autophagy Regulates PF Caused by Silica

We first observed the ultrastructure of alveolar epithelial cells in model mice lungs by TEM to study the potential mechanisms of autophagy in the development of silicosis. The results indicated that silica induced an increase in the number of autophagosome-like structures over time compared with that seen in saline controls (Figure 3A). Autophagosome numbers were significantly increased in TGF-β1-treated TC-1 cells compared to those in the control group, which was similar to the results of the in vivo experiment (Figure 3B). Moreover, we investigated the levels of a range of autophagy-related proteins in the silicosis mice lung tissue. The expression of Beclin1 and LC3 was upregulated in mouse lung tissues 14 and 28 days after silica exposure compared to the control mice, while no significant difference was observed in the day 7 model group (Figure 3C). Next, we evaluated the expression of autophagy in TGF-β1-treated TC-1 cells. The results indicated that the levels of LC3 and Beclin1 proteins were upregulated with increasing TGF-β1 concentrations and peaked at 10 ng/mL concentration (Figure 3D). Hence, the follow-up experiments were carried out at a concentration of 10 ng/mL of TGF-β1. TC-1 cells were then treated with 3-methyladenine, an autophagy inhibitor, or rapamycin, an autophagy inducer, together with TGF-β1. The results indicate that fibrosis marker proteins α-SMA and collagen I were increased in TGF-β1-treated TC-1 cells, and treatment with the 3-methyladenine further increased their expression. In contrast, rapamycin treatment reduced the α-SMA and collagen I levels in TC-1 cells (Figure 3E). These findings imply that autophagy protects lung epithelial cells from PF and attenuates silica-induced PF.

### 2.4. Inhibition of Akt3 Expression Promotes Autophagy in Lung Epithelial Cells and Regulates Fibrosis

In this study, we aimed to explore whether Akt3 regulates autophagy involved in PF. We first examined the expression of Akt3 and phospho-mTOR in a mouse model of silicosis. IHC and Western blot results showed that Akt3 levels were evidently elevated in the lung tissues of mice with silicosis compared to those in the control group mice, and phospho-mTOR protein levels were also increased (Figure 4A,B). In addition, with increased time after silica treatment, the expression of phospho-mTOR increased. Consistent with the results of the silicosis mouse model, in TC-1 cells treated with TGF-β1, we discovered that the mRNA expression of Akt3 increased in a dose-dependent manner, with the highest expression at 10 ng/mL TGF-β1 (Figure 4C). Next, we used 10 ng/mL TGF-β1 as a positive control. Additionally, to investigate the biological function of Akt3 in TGF-β1-treated TC-1 cells, (siRNA) transfection assays were performed. We constructed three siRNAs targeting Akt3 and evaluated the interference at the mRNA level. The siRNA with the strongest effect was selected for co-treating the TC-1 cells treated with TGF-β1 (Appendix A). Western blot analysis was used to determine the expression of phospho-mTOR, autophagy-related proteins Beclin1 and LC3α/β, and mesenchymal phenotype markers, α-SMA and collagen I. The research revealed that silencing of Akt3 promoted the level of Beclin1 and LC3 expression and reduced the expression of phospho-mTOR, α-SMA, and collagen I (Figure 4D). These results suggest that inhibition of Akt3 expression alleviates lung fibrosis via mTOR-mediated autophagy.

### 2.5. miR-29a-3p Targets Akt3 and Inhibits Its Expression

As previously mentioned, miR-29a-3p was decreased in silicosis mouse models both in vitro and in vivo. To investigate the mechanism by which upregulated miR-29a-3p inhibits PF, the online database TargetScanHuman 8.0 (https://www.targetscan.org/vert_80/, accessed on 10 March 2023) and miRDB (http://mirdb.org/, accessed on 10 March 2023) were used to predict the target gene of miR-29a-3p. The results showed that the 3′-UTR of Akt3 contains a miR-29a-3p-binding domain (Figure 5A). Next, the dual-luciferase reporter assay was used to confirm the relationship. The Akt3-WT or Akt3-MUT plasmid was co-transfected into TC-1 cells with or without the miR-29a-3p mimics. Our study demonstrated that the miR-29a-3p mimics significantly reduced the fluorescence intensity of the Akt3-WT group but not the Akt3-MUT group (Figure 5B). In addition, in TC-1 cells treated with TGF-β1, Western blot and qPCR data showed that overexpression of miR-29a-3p inhibited Akt3 expression. The miR-29a-3p mimics reduced the level of Akt3 (Figure 5C,D). Furthermore, after the transfection of miR-29a-3p inhibitor into cells, the expression of Akt3 protein and mRNA levels increased in TC-1 cells, similar to the response after TGF-β1 treatment (Figure 5E,F).

### 2.6. miR-29a-3p Regulates Autophagy via the Akt3/mTOR Axis to Attenuate Fibrosis In Vitro

Given the role of lung epithelial cell autophagy in PF and the downregulated expression of miR-29a-3p in lung fibrosis, we further investigated whether changes in miR-29a-3p could exert an anti-fibrotic effect by regulating lung epithelial cell autophagy. For this purpose, miR-29a-3p mimics were transfected into TC-1 cells to exogenously alter the expression of miR-29a-3p. The efficiency of the transfection was verified via qRT-PCR. Our results demonstrated that, compared to that in the control group, miR-29a-3p levels were clearly increased in the miR-29a-3p mimics group (Appendix A). Next, the miR-29a-3p mimics were transfected into TC-1 cells with or without TGF-β1 to explore the molecular mechanisms by which miR-29a-3p impacts the pathophysiology of PF. The results obtained using the mimics demonstrated that miR-29a-3p overexpression decreased Akt3 expression (Figure 5) and increased mTOR activity. Increased autophagy was supported by increased miR-29a-3p levels and the increased expression of Beclin1 and LC3α/β. More importantly, in TC-1 cells treated with TGF-β1, upregulation of miR-29a-3p expression reduced the levels of the fibrosis markers α-SMA and collagen I (Figure 6A). Moreover, IF data showed that overexpression of miR-29a-3p increased LC3 and inhibited α-SMA expression (Figure 6B). The results indicate that the Akt3/mTOR axis mediates the promotion of autophagy in pulmonary epithelial cells by miR-29a-3p, and overexpression of miR-29a-3p, at least partly, attenuates PF via Akt3/mTOR axis–mediated autophagy.

## 3. Discussion

Silicosis, a serious and irreversible type of PF caused by prolonged exposure to productive dust in occupational activities, is the biggest problem affecting occupational health in most low-income countries, including China [30]. Pulmonary fibrotic diseases, including silicosis, have been studied for a long time; however, their basic mechanism remains to be elucidated [31,32]. Many miRNAs play important roles in pulmonary fibrogenesis and could be a critical target for silicosis treatment [33,34]. This study aimed to determine the role of miR-29a-3p in silica-induced PF. We found that lung epithelial cells treated with TGF-β1 and silica-exposed mouse lung tissues both had considerably low levels of miR-29a-3p. More importantly, the data from bioinformatics analysis (GSE32538) show that miR-29a-3p levels were markedly decreased in patients with PF, indicating the underlying clinical significance of miR-29a-3p in the treatment of PF. We hypothesize that an increase in miR-29a-3p levels could alleviate PF by targeting Akt3 and suppressing the transduction of mTOR signaling pathway-induced lung epithelial cell autophagy and that the loss of miR-29a-3p expression aggravates the disease.

There is increasing research on miRNA dysregulation in multiple diseases, such as heart disease, cancer, and tissue fibrosis [35,36]. Current evidence indicates that miR-29 is reduced in many diseases associated with liver, heart, and lung fibrosis [26,27]. For example, the miR-29 family exerts anti-fibrotic effects by regulating fibrosis-related genes such as IGF1 and GTGF [37]. In addition, miR-29 is negatively regulated by TGF-β/Smad3 and has a potential therapeutic effect on bleomycin-induced pulmonary fibrosis [38]. However, whether miR-29a-3p alleviates PF induced by silica and the associated molecular mechanisms requires further investigation. In this study, TC-1 cells treated with TGF-β1 were transfected with miR-29a-3p mimics or inhibitor to investigate the role of miR-29a-3p in lung fibrosis. The results indicated that TGF-β1 elevated the expression of α-SMA and collagen I in TC-1 cells, whereas increased miR-29a-3p significantly decreased the expression of α-SMA and collagen I. In contrast, the reduction of miR-29a-3p further increased the expression of α-SMA and collagen I in TC-1 cells treated with TGF-β1. Thus, our results suggest that miR-29a-3p may act as a fibrosis inhibitor in silicosis.

Autophagy is important in maintaining normal lung function as well as in some lung diseases [39,40]. Studies have shown that autophagy is a cellular self-defense mechanism that can counteract lung damage caused by numerous factors, including SiO_2_ [41]. In addition, studies have found that epithelial cell dysfunction, including defective autophagy, is a central component of PF [10]. In the current study, we found that autophagy may contribute to the repair of silica-induced lung epithelial cell damage, thus exerting anti-fibrotic effects. Therefore, the autophagy of epithelial cells as a potential therapeutic target may help to alleviate the pathological development of PF. The levels of the autophagy-associated proteins Beclin1 and LC3α/β in mouse lung tissues and lung epithelial cells were examined. Our findings showed that TGF-β1-treated TC-1 cells and silicosis mice had considerably higher levels of Beclin1 and LC3. These results are consistent with previously reported findings [42]. TEM analysis showed a greater number of autophagosomes in the silicosis mouse model lung tissues over time. Consistent with these results, increased autophagosome numbers were also observed in TC-1 cells treated with TGF-β1 compared with that seen in controls. These results indicated that silica-induced autophagy. To determine how changes in lung epithelial cell autophagy regulate silica-induced PF, we performed a rescue experiment. For this, TGF-β1-induced TC-1 cells were treated with rapamycin, an autophagy activator, to activate autophagy and enhance autophagic flux. In TC-1 cells treated with TGF-β1, rapamycin decreased the protein levels of α-SMA and collagen I while increasing the expression of Beclin1 and LC3. On the contrary, additional treatment of the TGF-β1-treated TC-1 cells with 3-MA, an autophagy inhibitor, caused an additional increase in the levels of fibrosis-related proteins α-SMA and collagen I compared to their levels in TC-1 cells treated with TGF-β1 alone. These data demonstrate that enhancement of lung epithelial cell autophagy could attenuate silica-induced PF, whereas suppression of autophagy promotes the pathogenesis of silica-induced PF. The mechanism of autophagy in silicosis is complex and even contradictory; for example, blocking the autophagy of alveolar macrophages decreases apoptosis, which reduces silica-induced PF [43]. In the early stages of silicosis, the activation of autophagy plays a protective role for the organism, whereas in the later stages, with the excessive increase in autophagy, PF may further worsen. Exploring the exact mechanisms of autophagy in silicosis might help develop better treatments.

The Akt family mediates a wide range of cellular processes and is crucial in the development of many cancers and other disorders in humans [44,45]. For instance, the Akt3/mTOR signaling pathway regulates apoptosis induced by myocardial infarction via autophagy [46], indicating that Akt3 could regulate autophagy. However, whether Akt3 regulates PF requires further investigation. In this study, Akt3 was significantly upregulated in lung tissues of silicosis mice and TGF-β1-treated TC-1 cells, indicating that it might be important for the development of PF. To investigate whether inhibiting Akt3 affects PF, siR-Akt3 was designed to suppress Akt3. Comparatively to siR-NC transfected cells, Akt3 knockdown in TC-1 cells significantly reduced the expression of α-SMA and collagen I in the presence of TGF-β1 stimulation. As Akt3 regulates autophagy via the mTOR signaling pathway, we explored the mechanism by which Akt3 inhibits PF induced by silica. Our study suggested that inhibition of Akt3 reduced the level of phospho-mTOR, indicating blockage of the mTOR signaling pathway. In addition, silencing of Akt3 significantly increased the level of LC3 and Beclin1 induced by TGF-β1 in TC-1 cells, suggesting that Akt3 inhibition blocked the mTOR signaling pathway and increased autophagy, thereby reducing PF.

Published reports have indicated a close relationship between Akt3 signaling and various miRNAs targeting Akt3. For example, miR-181b-5p and miR-125b-5p regulate keratinocyte proliferation by targeting Akt3 [47]. In this study, the binding site of miR-29a-3p in the 3′-UTR of Akt3 was examined using the miRNA target gene prediction website. We used a dual-luciferase reporter assay to investigate the interaction between miR-29a-3p and Akt3. Notably, the miR-29a-3p attenuated levels of α-SMA and collagen I was rescued by downregulation of Akt3, indicating that the anti-fibrotic function of miR-29a-3p is exerted at least partially by targeting Akt3.

A major limitation of this study is the lack of sufficient silicosis population data and clinical patient research data, which limits the applicability of our results in a clinical setting. In addition, the development of disease is the result of multiple cell interactions in vivo. In this study, we demonstrated the anti-fibrotic effect of miR-29a-3p regulated autophagy only in vitro, but in vivo, evidence was lacking. That leads to a lack of a comprehensive understanding of pathogenicity. Subsequently, we will further explore the therapeutic mechanism of miR-29a-3p mediated autophagy in animal models of silicosis and its application as a target in the clinic.

In summary, our study explored a novel mechanism by which Akt3/mTOR signaling-based miRNAs regulate PF via autophagy. In accordance with earlier miRNA array investigations, our findings demonstrated a considerable downregulation of miR-29a-3p expression and an upregulation of autophagy in the lung tissues of mice exposed to silica and TC-1 cells treated with TGF-β1. Our experimental results demonstrate that miR-29a-3p decreases the TGF-β1-induced elevation of fibrosis marker proteins by increasing autophagy in vitro and shows a distinct anti-fibrotic effect. Akt3, an important upstream gene in the mTOR signaling pathway, was identified as a direct target of miR-29a-3p. The study revealed that the suppression of PF by miR-29a-3p was at least partially dependent on the direct regulation of Akt3 expression. Our findings elucidate the mechanism of the pro-autophagic and anti-fibrotic effects of miR-29a-3p in silicosis and indicate that the miR-29a-3p/Akt3/mTOR axis plays a crucial role in silica-induced lung fibrosis. Therefore, our findings suggest that miR-29a-3p may someday represent a promising target for the treatment of silicosis. However, further research is necessary before its clinical application.

## 4. Materials and Methods

### 4.1. Bioinformatics Analysis

Microarray expression profiling data were obtained from the Gene Expression Omnibus (GEO, https://www.ncbi.nlm.nih.gov/geo/, accessed on 10 March 2023) with accession numbers GSE45789 and GSE32538. The GSE45789 dataset contains the miRNA profiles of lung tissues from three mice with PF and three healthy controls. The GSE32538 dataset includes miRNA profiles of lung samples from 106 patients with idiopathic pulmonary fibrosis (IPF) and 50 non-diseased controls. The statistical analysis for the miRNA profiling of GSE45789 and GSE32538 was based on the GPL7723 and GPL8786 platforms, respectively. GEO2R online analysis was used to define the differential expression of miRNAs (*p* < 0.05). DAVID 2021 (https://david.ncifcrf.gov, accessed on 10 March 2023) was used for the Kyoto Encyclopedia of Genes and Genomes (KEGG) pathway enrichment analysis. When the *p*-value was less than 0.05, the results of the KEGG enrichment analysis were considered significant. The online database miRDB (http://mirdb.org, accessed on 10 March 2023) was used to identify the target genes of miR-29a-3p.

### 4.2. Animals

Male C57BL/6J mice aged 6 weeks (23 ± 2 g) were purchased from Pekin Huafukang Bioscience Co., Ltd. (SCXY 2019-0008, Beijing, China) and housed in a 12-h light/dark cycle environment at a temperature of 25 ± 2 °C. The mice were allowed to acclimate for 1–2 weeks and maintained on commercial standard chow with free access to distilled water before silica treatment. All protocols of the animal experiment were reviewed and approved by the North China University of Science and Technology Ethics Committee (No. 2021-007).

### 4.3. Silica-Induced PF Mouse Model

Mice were randomly divided into four groups for days 7, 14, 28, and the control group (n = 8 animals per group). The mice were injected intraperitoneally with 1.25% tribromoethanol (Jitian Bio, Beijing, China) before silica treatment. The silicosis mice group was treated by intratracheal instillation of a one-time dose of 10 mg/mouse (0.05 mL) silica suspension (Sigma Aldrich, Burlington, VT, USA). Mice in the control group were given equal amounts of saline intratracheally. On days 7, 14, and 28, mice were anesthetized and euthanized. Paraformaldehyde (Biosharp, Beijing, China) was used to fix the right lower lung for histological examination. Additional lung tissues were taken out and promptly stored at −80 °C for later investigation.

### 4.4. Histopathological and Immunohistochemistry (IHC) Analysis

The lungs were immediately removed, and the right lung tissue was fixed for 48 h with 4% paraformaldehyde. The tissues were then dehydrated in a graded ethanol series and embedded in paraffin. For pathological analysis, 4 μm thick sections were stained with hematoxylin and eosin (HE). Morphological evaluation of the specimens by Masson’s trichrome staining (Leagene Biotechnology, Beijing, China) was performed to determine the distribution of fibrous collagen in lung tissue. The sections were then examined under a microscope (Olympus Optical Co., Ltd., Tokyo, Japan).

IHC was performed to assess the expressions of Akt3 in the lung tissue of the mouse model. In brief, the dewaxed tissue sections were subjected to antigen retrieval for 30 min at 95 °C. Slides were then incubated overnight with primary antibodies against Akt3 (1:100 dilution; Wanleibio, Shenyang, China) at 4 °C. The slides were then washed and incubated with an anti-rabbit secondary antibody (Thermo Fisher, Waltham, MA, USA) at 37 °C. The slides were then examined using a microscope (Olympus Optical Co., Ltd., Tokyo, Japan), and images were captured.

### 4.5. Cell Culture, Plasmid Construction, and Transfection

TC-1 mice lung epithelial cells were provided by the Stem Cell Bank of the Chinese Academy of Sciences (Shanghai, China). The TC-1 cells within the 10th passage were used for the subsequent experiments and were cultivated in Dulbecco’s modified Eagle medium (Gibco, Brooklyn, NY, USA) containing 10% fetal bovine serum (Gibco, Brooklyn, NY, USA) at 37 °C with 5% CO_2_ in an incubator and were exposed to varying concentrations (0, 1, 5, and 10 ng/mL) of TGF-β1 for 48 h to construct cell models. TC-1 cells were then treated with 10 mM 3-methyladenine (Sigma Aldrich, Burlington, VT, USA) or 20 nM rapamycin (Sigma Aldrich, Burlington, VT, USA) to inhibit and activate autophagy, respectively. The miRNA mimics/inhibitors and small interfering RNA (siRNAs) were constructed by GenePharma Co., Ltd. (Shanghai, China), and the sequences are listed in Appendix A. When TC-1 cells reached 60% confluence in 6-well plates, they were transiently transfected with 30 nM miR-29a-3p mimics or inhibitor or 50 nM siR-Akt3 using HighGene transfection reagent (ABclonal, Wuhan, China).

### 4.6. RNA Extraction and Quantification

TRIzol (Invitrogen, Waltham, MA, USA) reagent was used to isolate total RNA from lung tissues and harvested TC-1 cells. A NanoDrop 2000 spectrophotometer (Thermo Fisher, Waltham, MA, USA) was used to validate the quality and concentration of RNA. Subsequently, cDNA was obtained by reverse transcription primers using the PrimeScript™ RT reagent kit (Takara, Shiga, Japan) for mRNA and Mir-X miRNA First-Strand Synthesis Kit (Takara, Shiga, Japan) for miRNA, following the manufacturer’s protocol. cDNA amplification was conducted by TB Green^®^ Premix Ex Taq™ II (Takara, Shiga, Japan). Quantitative reverse transcription polymerase chain reaction (qRT-PCR) was conducted using ABI 7900HT Real-Time PCR System (Applied Biosystems, USA). U6 and GAPDH were selected as an internal control for the miRNA and mRNA, respectively. The 2^−ΔΔCT^ (ΔΔCT = ΔCT(experimental) − ΔCT(negative control)) method was employed to determine the level of miR-29a-3p and Akt3. Specific primers for miR-29a-3p were designed by GenePharma Co., Ltd. (Shanghai, China). The primer pair sequences are listed in Appendix A.

### 4.7. Western Blotting

Proteins were extracted from lung tissues and cells using a radio-immunoprecipitation assay buffer containing protease inhibitors. The BCA protein assay kit (Thermo Scientific, Shanghai, China) was used to determine the protein concentration as per the manufacturer’s protocol. A total of 15 μg protein extracts were electrophoresed in 10% sodium dodecyl sulfate-polyacrylamide gels and transferred to NC membranes (PALL, Cortland, NY, USA) via electrophoretic transfer. After blocking with 5% non-fat milk for 2 h at room temperature (25 ± 2 °C), the NC membranes were incubated overnight at 4 °C with the following specific primary antibody: anti-phospho-mTOR (1:1000; Affinity, Cincinnati, USA), anti-collagen I (1:1000; Affinity, Cincinnati, OH, USA), anti-alpha-SMA (1:1000; Affinity, Cincinnati, OH, USA), anti-Akt3 (1:1000; Wanleibio, Shenyang, China), anti-Beclin1 (1:1000; Wanleibio, Shenyang, China), anti-LC3α/β pAb (1:1000; MBL, Nagoya, Japan), and anti-GAPDH (1:1000; Affinity, Cincinnati, OH, USA). The membranes were washed three times with TBS-T and then incubated with secondary antibodies corresponding to the host species of the primary antibodies: goat anti-rabbit IgG (Affinity, Cincinnati, OH, USA) or goat anti-mouse IgG (Affinity, Cincinnati, OH, USA) at a dilution ratio of 1:5000 for 1.5 h at room temperature (25 ± 2 °C). After washing three times, the signals were measured using enhanced chemiluminescence (ECL) detection reagent (Applygen, Beijing, China). ImageJ (version 1.51) was used to analyze the densities of target proteins.

### 4.8. Immunofluorescence

TC-1 cells transfected with miR-29a-3p were treated with 10 ng/mL of TFG-β1 for 24 h and then fixed with 4% paraformaldehyde for 30 min, washed 3 times with PBS, and serum blocked for 10 min. Then the cell was incubated overnight at 4 °C with the following specific primary antibody: anti-alpha-SMA (1:100; Affinity, Cincinnati, OH, USA), anti-LC3α/β pAb (1:100; MBL, Nagoya, Japan), washed 3 times with PBS and then incubated with fluorescence secondary antibodies corresponding to the host species of the primary antibodies donkey anti-rabbit IgG (Invitrogen, Waltham, MA, USA) or goat anti-mouse IgG (ABclonal, Wuhan, China) at a dilution ratio of 1:100 for 1.5 h at 37 °C. DAPI (Abcam, Boston, MA, USA) was used to stain the nuclei, then examined using a fluorescence microscope (Olympus Optical Co., Ltd., Tokyo, Japan), and images were captured.

### 4.9. Detection of Autophagosomes Using Transmission Electron Microscopy (TEM)

Lung tissue (three mice per group) and pulmonary epithelial cells treated with TGF-β1 were fixed with 2.5% glutaraldehyde and then post-fixed in 1% OsO4 for 1 h. Next, the samples were dehydrated in acetone and embedded in Epon 812 (SPI Supplies, 02635-AB). Uranyl acetate/lead citrate was used to stain thin sections, which were then examined using the Hitachi H-7650 transmission electron microscope (Tokyo, Japan) at 80 kV.

### 4.10. Dual-Luciferase Reporter Assay

To investigate the interaction between miR-29a-3p and Akt3, we cloned 3′-UTR segments of Akt3 containing binding sites of miR-29a-3p into a psiCHECK-2 (GenePharma, Shanghai, China) vector named Akt3 wild-type (Akt3-WT) plasmid and constructed a mutant plasmid (Akt3-MUT). TC-1 cells were co-transfected with Akt3-WT or Akt3-MUT together with miR-29a-3p mimics or mimics control using HighGene transfection reagent (ABclonal, Wuhan, China). A dual-luciferase assay system (Promega, Madison, WI, USA) was used to detect firefly luciferase activities according to the manufacturer’s protocol and adjusted by Renilla luminescence.

### 4.11. Statistical Analysis

Statistical Package for Social Sciences (SPSS) v21.0 (SPSS Inc., Chicago, IL, USA) was used for data analyses. The differences between treatment groups and their respective controls were compared via independent sample *t*-tests. The difference between groups for different treatments was compared using a one-way analysis of variance. When two-tailed *p* < 0.05, the result was considered statistically significant.

## 5. Conclusions

These results suggest that miRNA in lung epithelial cells plays a crucial role in the pathology of silicosis. Via bioinformatic analysis, we identified an association between miR-29a-3p and SiO2-induced PF. The results reveal that miR-29a-3p regulates autophagy in pulmonary epithelial cells by targeting Akt3, thereby alleviating PF and highlighting the potential of miR-29a-3p as a target for the treatment of this disease.

## Figures and Tables

**Figure 1 ijms-24-11440-f001:**
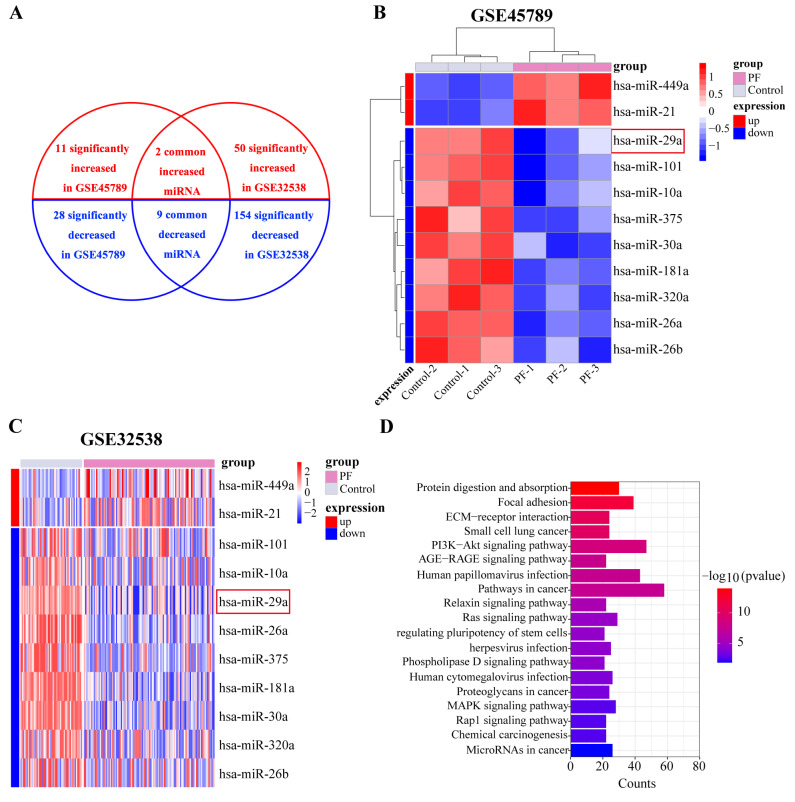
miR-29a expression is significantly downregulated in fibrotic lung tissue, as demonstrated by bioinformatics analysis. (**A**) Venn diagram showing the highly expressed, lowly expressed, and overlapping miRNAs in GSE45789 and GSE32538. (**B**,**C**) Expression of different miRNAs is represented by heat maps and sample clustering; Red boxes indicate decreased miR-29a. (**D**) KEGG pathway enrichment of the predicted target genes of miR-29a-3p.

**Figure 2 ijms-24-11440-f002:**
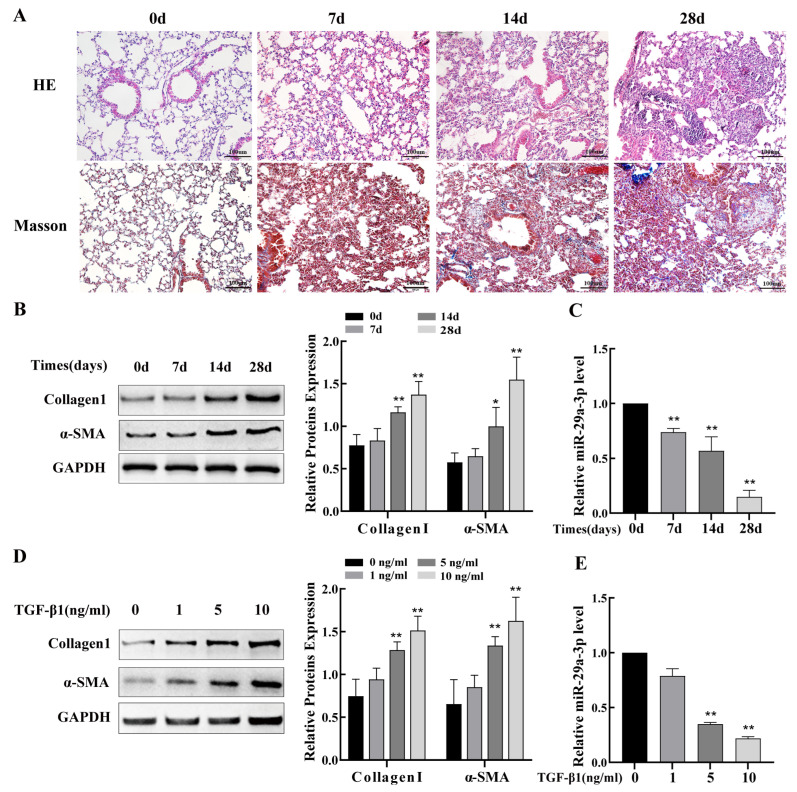
miR-29a-3p expression was downregulated both in vitro and in vivo in the silica-induced pulmonary fibrosis model. (**A**) Histological changes and collagen deposition in the lung tissues of silicosis model mice were observed upon hematoxylin and eosin (HE) and Masson staining. (**B**) The expression of α-SMA and collagen I in the lung tissues of mice administered 50 mg/kg silica for 7, 14, and 28 days measured using Western blot compared with that in the control group. (**C**) miR-29a-3p levels were measured using qRT-PCR in the lung tissues of mice administered silica. (**D**) Western blot analysis of collagen I and α-SMA levels in TC-1 cells treated with various concentrations of TGF-β1. (**E**) miR-29a-3p levels in TC-1 cells treated with various concentrations of TGF-β1, as determined via qRT-PCR. Data are presented as the mean ± SD from at least three independent experiments. * indicates *p* < 0.05 vs. the control group, and ** indicates *p* < 0.01 vs. the control group.

**Figure 3 ijms-24-11440-f003:**
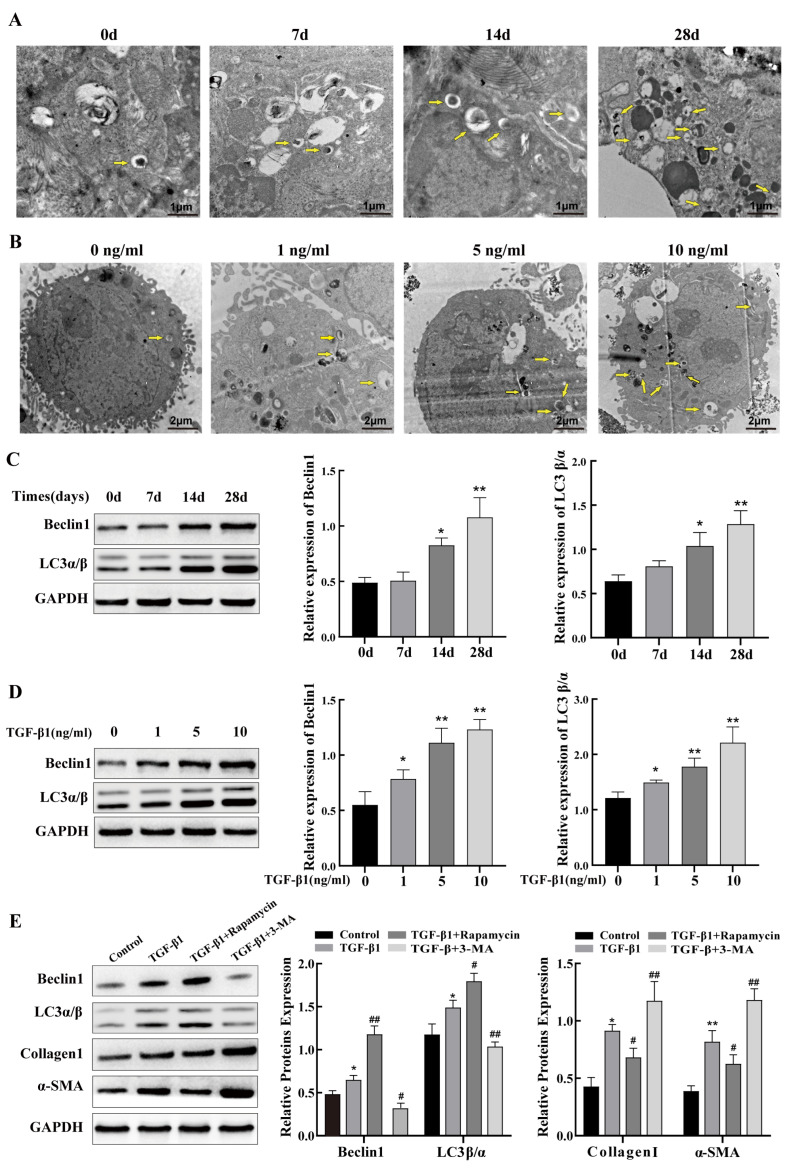
Autophagy regulates silica-induced pulmonary fibrosis. (**A**,**B**) Representative electron microscopy images from mouse lung tissues exposed to 50 mg/kg silica for 0, 7, 14, and 28 days and TC-1 cells treated with different concentrations (0, 1, 5, and 10 ng/mL) of TGF-β1 for 48 h; yellow arrows indicate autophagosomes. (**C**) Western blot analysis of proteins associated with autophagy Beclin1 and LC3-α/β level in mouse lung tissues exposed to silica. (**D**) Western blot analysis of proteins associated with autophagy Beclin1 and LC3-α/β level in TGF-β1-treated TC-1 cells. (**E**) Western blot results of Beclin1 and LC3α/β as well as fibrosis markers, α-SMA and collagen I, in TC-1 cells treated with autophagy inhibitor (3-MA) and inducer (rapamycin) together with 10 ng/mL TGF-β1. Data are presented as the mean ± SD from at least three independent experiments; * indicates *p* < 0.05 vs. the control group, ** indicates *p* < 0.01 vs. the control group, # indicates *p* < 0.05 vs. the TGF-β1 group, and ## indicates *p* < 0.01 vs. the TGF-β1 group.

**Figure 4 ijms-24-11440-f004:**
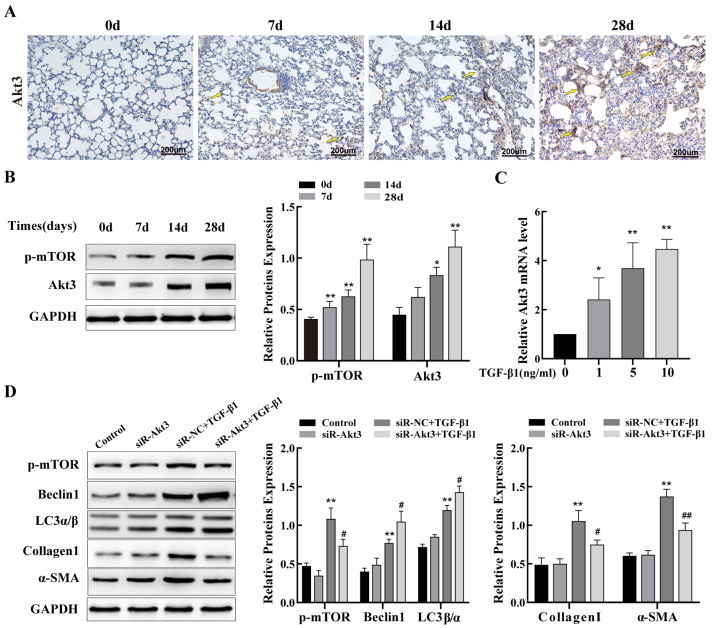
Inhibition of Akt3 expression promotes autophagy in lung epithelial cells and ameliorates fibrosis by activating mTOR. (**A**) Representative images of Akt3 immunohistochemistry in lung tissue of mice with silicosis; Yellow arrows indicate positive expression of Akt3. (**B**) Western blot analysis of Akt3 and phospho-mTOR levels in lung tissues of mice with silicosis. (**C**) qRT-PCR analysis of Akt3 levels from TC-1 cells treated with various doses of TGF-β1. (**D**) The expression of phospho-mTOR, autophagy-related proteins LC3α/β and Beclin1, and fibrosis markers, α-SMA and collagen I, were verified using Western blot in TC-1 cells transfected with 50 nM of siR-Akt3 or siR-NC before treatment with 10 ng/mL of TGF-β1. Data are presented as the mean ± SD from at least three independent experiments; * indicates *p* < 0.05 vs. the control group, ** indicates *p* < 0.01 vs. the control group, # indicates *p* < 0.05 vs. the siR-NC plus TGF-β1 group, and ## indicates *p* < 0.01 vs. the siR-NC plus TGF-β1 group.

**Figure 5 ijms-24-11440-f005:**
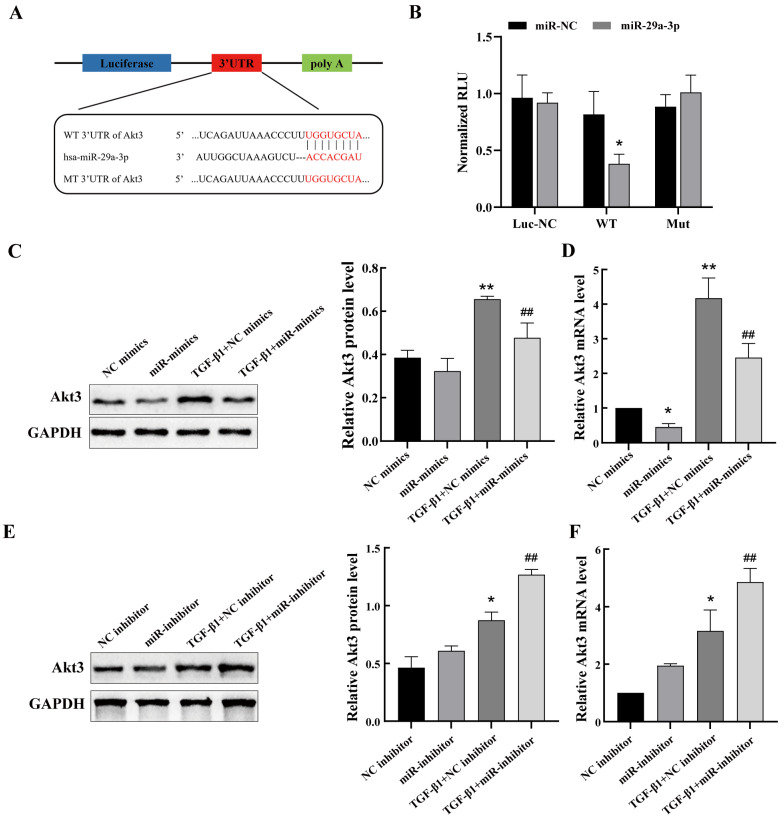
miR-29a-3p targets Akt3 and inhibits its expression. (**A**) The location of the miR-29a-3p target site in the Akt3 3′ UTR. The red region denotes the binding sequences between miR-29a-3p and the 3′-UTR of Akt3. (**B**) The interaction of Akt3 and miR-29a-3p in TC-1 cells was validated using a dual-luciferase reporter assay. The intensity of luciferase activity is expressed as the ratio of firefly/Renilla activity. (**C**,**D**) Western blot and qPCR results of Akt3 levels in TC-1 cells transfected with miR-29a-3p mimics before treatment with 10 ng/mL TGF-β1. (**E**,**F**) Western blot and qPCR results of Akt3 levels in TC-1 cells transfected with miR-29a-3p inhibitor before treatment with 10 ng/mL TGF-β1. Data are presented as the mean ± SD from at least three independent experiments; * indicates *p* < 0.05 vs. the control group, ** indicates *p* < 0.01 vs. the control group, ## indicates *p* < 0.01 vs. the TGF-β1 plus NC mimics/inhibitor group.

**Figure 6 ijms-24-11440-f006:**
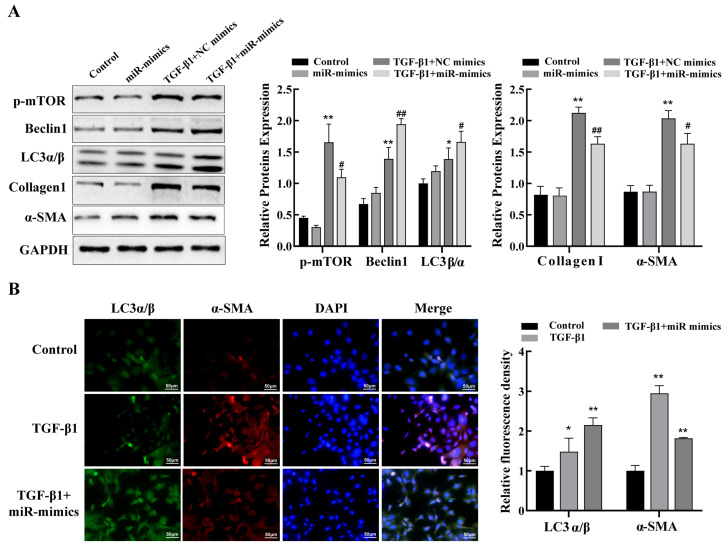
miR-29a-3p regulates autophagy via the Akt3/mTOR axis to attenuate TGF-β1–induced fibrosis in vitro. (**A**) Transfection of miR-29a-3p mimics into TC-1 cells followed by treatment with 10 ng/mL of TGF-β1. The expression of phospho-mTOR, autophagy-associated proteins LC3α/β and Beclin1, as well as fibrosis markers α-SMA and collagen I in TC-1 cells, were detected via Western blot. (**B**) Immunofluorescence results of LC3α/β and α-SMA in TC-1 cells transfected with miR-29a-3p mimics before treatment with 10 ng/mL TGF-β1. Data are presented as the mean ± SD from at least three independent experiments; * indicates *p* < 0.05 vs. the control group, ** indicates *p* < 0.01 vs. the control group, # indicates *p* < 0.05 vs. the TGF-β1 plus NC mimics group, and ## indicates *p* < 0.01 vs. the TGF-β1 plus NC mimics group.

## Data Availability

Not applicable.

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
