# Peer review of "miR-29a-3p Regulates Autophagy by Targeting Akt3-Mediated mTOR in SiO2-Induced Lung Fibrosis"

_ijms, 2023, doi:10.3390/ijms241411440_

Round 1

Reviewer 1 Report

The article developed is part of the field of knowledge about new methods of understanding and managing pulmonary fibrosis. The mechanisms that involve the phenomena that determine the progression of the structural and functional changes in pulmonary fibrosis are mostly unknown. Therefore, the utility of the present study is fundamental, taking into account the severity of the disease and the absence of effective treatments.

In general, the article is well developed, the techniques used are correct and no grammatical or form errors are reported. The graphics and images help to understand the research. The references used are adequate in number and form.

However, it is recommended to reformulate the structure of the text. The materials and methods section should be prior to the results, in order to facilitate the understanding of the tests carried out and the results presented.

Author Response

Dear Professor and Reviewer:

We sincerely appreciate your valuable feedback on our manuscript. For the text structure you mentioned, we have arranged it according to the template of your journal, and the methods and materials follow the results. In addition, we have made revisions to the manuscript based on the comments of the reviewers and carefully proofread it to reduce printing, grammar, and bibliographic errors. We hope you can review it.

Sincerely you,

Reviewer 2 Report

This paper explores the role of MicroRNAs (miRNAs) in the development of silicosis, specifically focusing on the relationship between miR-29a-3p, autophagy, and pulmonary fibrosis. The findings the authors presented shed light on the molecular mechanisms underlying silicosis and suggest a potential therapeutic strategy for this disease. While this study primarily focuses on the role of miR-29a-3p and Akt/mTOR pathway in autophagy and fibrosis, it is important to consider potential off-target effects or additional molecular pathways that may be involved.

Comments:

Introduction

While the current study provide background information on autophagy and miRNAs, it would be helpful to explicitly highlight the gap in knowledge or the research question that the study aims to address. This will help readers understand the importance and novelty of the research.

Results

1.       Connect the findings on miR-29a-3p's downregulation and its predicted target genes with the research question or objective of this study. This will help readers understand the relevance and significance of miR-29a-3p in the context of the current research.

2.       Consider providing more specific details about the observed parenchymal alterations and inflammatory cell infiltration. Additionally, describe the time points at which these alterations were observed (7 days, 14 days, and 28 days), which would provide a clearer understanding of the progression of PF.

3.       Instead of referring to supplemental figures (Fig. S1 and Fig. S2), consider summarizing the key findings related to the expression of autophagy-related proteins, such as Beclin1 and LC3, in the silica-exposed lung tissue and TGF-β1-treated TC-1 cells. Clearly state how the expression levels of these proteins changed in response to silica exposure or TGF-β1 treatment.

4.       Need to clearly state the dose-dependent increase in Akt3 mRNA expression in TC-1 cells treated with TGF-β1. Specify the range of TGF-β1 concentrations used and how they corresponded to the observed increase in Akt3 expression.

Discussion

1.       The authors could provide additional insights into the limitations and potential biases associated with the experimental design or methodology employed. Additionally, suggest possible avenues for future research, such as in vivo studies using animal models or clinical studies involving human samples, to validate the findings and further elucidate the mechanisms of autophagy in silicosis.

2.       The current study mention that previous studies highlighting the association between miR-29 and various diseases. To strengthen the argument, consider providing a brief summary or comparison of these studies, particularly those related to fibrosis or lung diseases. This would help situate the current findings within the existing body of literature and emphasize the novelty or significance of this research.

Methods

1.       The study exclusively used male mice, which may limit the generalizability of the findings to both genders.

2.       Specify the passage number or range of TC-1 cells used in the study.

3.       There are several types of miR-29 (miR-29a, b,c). If applicable, provide details on the design of the specific primers for miR-29a-3p by GenePharma Co. Ltd. Include the primer sequences.

4.       Include the formula or equation used for the 2-ΔΔCT method.

Author Response

Dear professor and reviewer:

Thank you very much for your comments for the manuscript. We revised the manuscript in accordance with your comments, and carefully proof-read the manuscript to minimize typographical, grammatical, and bibliographical errors. The attachment is our description on revision according to the reviewers’ comments.

Sincerely yours,

Round 2

Reviewer 2 Report

No more comments.